# Light Quality Affected the Growth and Root Organic Carbon and Autotoxin Secretions of Hydroponic Lettuce

**DOI:** 10.3390/plants9111542

**Published:** 2020-11-11

**Authors:** Chengbo Zhou, Yubin Zhang, Wenke Liu, Lingyan Zha, Mingjie Shao, Baoshi Li

**Affiliations:** 1Institute of Environment and Sustainable Development in Agriculture, Chinese Academy of Agricultural Sciences, Beijing 100081, China; 82101181045@caas.cn (C.Z.); 82101175084@caas.cn (Y.Z.); 82101161036@caas.cn (L.Z.); 82101182120@caas.cn (M.S.); 82101186224@caas.cn (B.L.); 2Key Lab of Energy Conservation and Waste Management of Agricultural Structures, Ministry of Agriculture and Rural Affairs, Beijing 100081, China

**Keywords:** botany, plant factory, LED, light quality, autotoxicity

## Abstract

Light is a crucial environmental signal and photosynthetic energy for plant growth, development, and primary and secondary metabolism. To explore the effects of light quality on the growth and root exudates of hydroponic lettuce (*Lactuca sativa* L.), white LED (W, control) and four the mixtures of red (R) and blue (B) LED with different R/B light intensity ratios (R/B = 2, 2R1B; R/B = 3, 3R1B; R/B = 4, 4R1B; and R/B = 8, 8R1B) were designed. The results showed that the biomass of lettuce under 8R1B and W treatments was higher than that under other light quality treatments. The photosynthetic rate (*P*n) under red and blue light was significantly higher than that of white light. Total root length, root surface area, and root volume were the highest under 8R1B. 4R1B treatment significant increased root activity by 68.6% compared with W. In addition, total organic carbon (TOC) content, TOC content/shoot dry weight, TOC content/root dry weight, and TOC content/root surface area were the highest under 4R1B. Moreover, 8R1B treatment reduced the concentration of benzoic acid and salicylic acid, and the secretion ability of benzoic acid and salicylic acid by per unit root surface area and accumulation by per unit shoot dry weight. In addition, 2R1B and 3R1B reduced the secretion ability of gallic acid and tannic acid by per unit root surface area and accumulation by per unit shoot dry weight. In conclusion, this study showed that the secretion of autotoxins could be reduced through the mediation of red and blue light composition of LEDs in a plant factory. In terms of autotoxin secretion reduction efficiency and yield performance of lettuce, 8R1B light regime is recommended for practical use.

## 1. Introduction

Plant factories have lately attracted more attention and developed greatly all over the world, particularly in East Asia countries. It usually uses automatic hydroponic circulation system of nutrient solution for plant cultivation, which can effectively improve nutrient utilization efficiency and reduce the discharge of waste nutrient solution for environmental protection [1,2]. However, the growth of plants, especially hydroponic vegetables, is often inhibited under the condition of hydroponics with a recirculating nutrient solution system. This allelopathy phenomenon was ascribed to the accumulation of autotoxins secreted by roots in the nutrient solution after multiple cropping [3,4].

Accumulation of phenolic acids in rhizosphere can inhibit growth and reduce plant yield, causing continuous cropping obstacles through allelopathic impacts [5]. The ability to secrete a great many compounds into the rhizosphere is one of the most significant metabolic characteristics of plant roots [6]. Phenolic acids are the main substances that produce autotoxicity in plant root exudates, and also belong to a group of important secondary metabolites that function during plant physiological processes [7]. Previous studies noted that excessive accumulation of root exudates inhibited cucumber growth, decreased net photosynthetic rate and stomatal conductance, and enhanced membrane lipid peroxidation [3,4].

Besides renewal of nutrient solution, some methods were developed to remove autotoxins accumulated in the nutrient solution of hydroponic culture. Lee et al. [8] found that activated carbon could effectively remove the phytotoxic substance accumulated in nutrient solution and reduce the inhibition on lettuce growth, but its removal ability is limited since high-dose activated carbon treatment could not eliminate the harmful impact of these phytotoxic substances on plant growth. The nano-TiO_2_ photocatalytic method effectively removed the autotoxins in the nutrient solution and increased the yield of plants [9,10], and also reduced the content of available Fe, Mn, and Zn [11]. Furthermore, Talukder et al. [12] used the electrodegradation method to significantly reduce the concentration of benzoic acid in the nutrient solution and increase the yield of strawberry, but the operation of this method is complicated and the investment is high.

Recently, light-emitting diodes (LEDs) with a variety of advantages (low energy consumption, adjustable light intensity and wavelength, etc.) were fully used to study the responses of growth, development, and morphogenesis of horticultural crops to various light environments [13,14]. Goins et al. [15] found that wheat under a white fluorescent lamp had the highest yield, but the biomass and yield under red and blue light were similar to those under white light, so the combination of red and blue light could replace continuous spectrum for plant cultivation using artificial light. Moreover, previous studies indicated that red light remarkably increased starch and soluble sugar content in plants by inhibiting the output of photosynthates from leaves [16]. Blue light regulated the physiological processes of chlorophyll formation, chloroplast development, and stomatal opening [17,18]. Additionally, combination of red and blue light improved photosynthetic rate, carbohydrate accumulation, and plant quality [19,20]. The light quality had an important influence on the growth and development of plant roots. Light quality could not only regulate the root formation, but also affect the root activity and the secondary metabolism of roots [21,22]. In addition, we recently reported that the secretion of autotoxins in lettuce was influenced by red and blue (R/B = 4:1) light intensity, and fewer autotoxins were secreted by roots at 200 μmol·m^−2^·s^−1^ [23]. Therefore, we suspect that the changes in LED light quality also have some potential functions to reduce root exudates.

Lettuce (*Lactuca sativa* L.) as a global vegetable with rich nutrients such as protein and vitamins is widely cultivated in facilities and gardening [24], and it is known to be sensitive to phytotoxic substances [3]. There are many studies on the effects of LED light quality on the growth and development of lettuce [25,26]. Moreover, nutrients and other conditions on the allelopathic effects of plants also are reported [27,28,29]. However, little information is available on how and whether LED light quality impacts root exudates of hydroponic leafy vegetables in a plant factory. The aim of this study was to determine the effects of light qualities with different R/B ratios on the (1) growth and yield, (2) organic carbon secretion, and (3) secretion of five autotoxins (benzoic acid, salicylic acid, ferulic acid, gallic acid, and tannic acid) of hydroponic lettuce. We endeavor to find out whether autotoxins in the nutrient solution can be reduced by mediating the light quality, so as to increase nutrient solution utilization efficiency and reduce the probability of continuous cropping obstacles in plant factories.

## 2. Materials and Methods

### 2.1. Plant Materials and Growth Condition

Lettuce (*Lactuca sativa* L. cv. ”Tiberius”) seeds were sown in sponge cubes (2.5 × 2.5 × 2.5 cm) and germinated under 200 μmol·m^−2^·s^−1^ irradiance provided by cold-white LED for 15 days in an environment-controlled plant factory. Environmental conditions in the experiment were set at 23 ± 2 °C, 60% relative humidity (RH) and 419 ± 3 μmol·mol^−1^ CO_2_ level. After the second leaf was fully expanded, the seedling roots were washed 3 times with deionized water and the seedlings were randomly transplanted into hydroponic pots containing 4 L of nutrient solution. There were 4 hydroponic pots for each treatment and 4 plants were planted in each pot. Lettuce plants were cultured in Hoagland’s solution containing 4 mM Ca(NO_3_)_2_·4H_2_O, 6 mM KNO_3_, 1 mM NH_4_H_2_PO_4_, 2 mM MgSO_4_·7H_2_O, 71 μM Fe–EDTA–Na_2_, 46 μM H_3_BO_3_, 9.6 μM MnSO_4_·4H_2_O, 0.8 μM CuSO_4_·5H_2_O, and 0.07 μM (NH_4_)_6_Mo_7_O_24_·4H_2_O (EC: 1.422 mS·cm^−1^, pH: 5.8).

### 2.2. Light Treatment

In the LED experiment, lettuce was irradiated 16 h per day (24 h) by cold-white LED (W) and combination of red and blue LED (RB). The size of the LED light panel is 50 × 50 cm (Shenzhen Huihao Optoelectronic Co. Ltd., Shenzhen, China). The LED light provided red with peak wavelength of 655 nm and blue with peak wavelength of 437 nm. The LED light panel was installed 25 cm above the hydroponic pots. The photon flux density (PFD) and light spectra (Table 1 and Figure 1) were monitored using a spectroradiometer (Avaspec-ULS2048, Avantes, Apeldoorn, The Netherlands). The probe of the spectroradiometer was placed in the center just below the lamp panel at the canopy level. Five light quality treatments based on different R/B light intensity ratios were designed in the experiment: R/B = 2 (2R1B), R/B = 3 (3R1B), R/B = 4 (4R1B), R/B = 8 (8R1B), and white light (W, control).

### 2.3. Sampling and Measurement Methods

Four lettuce plants of each treatment were randomly sampled for the measurements of leaf area, shoot fresh weight, root fresh weight, shoot dry weight, and root dry weight on the 15th day after treatment. Leaf area was measured using an area meter (LI-3100, Li-Cor, Lincoln, NE, USA). Dry weight was determined after drying at 80 °C for 48 h.

#### 2.3.1. Net Photosynthetic Rate

The photosynthetic rate (*P*n) of the fully expanded 2nd leaf from the top was measured under growth light intensity using a portable photosynthesis system (LI-6400, Li-Cor, Lincoln, NE, USA). Leaf temperature and CO_2_ concentration in the leaf chamber were 22 °C and 400 μmol·mol^−1^, respectively.

#### 2.3.2. Root Morphology

Root morphology was measured twice. For the first time, 15 days after pot cultivation in the nutrient solution under different light conditions, lettuce plants were randomly sampled and divided into shoots and roots. Fresh roots were washed with deionized water and scanned using an Epson Perfection V850 photoscanner (Epson, Nagano, Japan), and the images were analyzed by WinRHIZO Pro software (version 2019, Regent Co. Ltd., Quebec, Canada). Total root length, root surface area, and root volume were determined. For the second time, 3 days after cultivation in deionized water, lettuce plants were divided into shoots and roots; fresh roots were determined for root morphology.

#### 2.3.3. Root Activity

Root activity was analyzed by the triphenyl tetrazolium chloride (TTC) method [30]. TTC is a chemical that is reduced by dehydrogenases, mainly succinate dehydrogenase, when added to a tissue. The dehydrogenase activity is regarded as an index of the root activity. In brief, 0.5 g fresh root was immersed in 10 mL of equally mixed solution of 0.4% (*w/v*) TTC and 0.1 M K-phosphate buffer (pH 7.5), and kept in the dark at 37 °C for 2 h. Subsequently, 2 mL of 1 M H_2_SO_4_ was added to stop the reaction with the root. The root was dried with filter paper and then extracted with ethyl acetate. The red extractant was transferred into a volumetric flask to reach 10 mL by adding ethyl acetate. The absorbance of the extract at 485 nm was recorded. Root activity was expressed as TTC reduction intensity: root activity = amount of TTC reduction (mg)/fresh root weight (g) × time (h).

#### 2.3.4. Collection and Determination of Total Organic Carbon

Taking out lettuces on the 15th day after treatment, the volume of the residual nutrient solution in hydroponic pot was measured by measuring cylinder. The completely mixed nutrient solution was filtered by a 0.45 μm filter membrane and stored at −80 °C for later analyses. Total organic carbon was determined using a TOC analyzer (Multi N/C 3100, Analytik Jena AG).

#### 2.3.5. Collection and Determination of Autotoxins in Deionized Water

Four lettuce plants of each treatment were randomly selected from four hydroponic pots. Roots were washed three times with deionized water. Then plants were placed in a 100 mL triangular bottle containing 120 mL deionized water. The triangular bottle was covered with aluminum foil paper to shade the light, and then plants were placed under the original light treatment. After 3 days, lettuce was taken out and roots were washed three times with deionized water, then the rinse solution was transferred to the triangle bottle containing residual solution and the deionized water was added to a volume of 120 mL in the triangle bottle. The solution was then filtered using 0.45 μm filter membrane and stored at −80 °C for later analyses.

The concentration of benzoic and salicylic acid was measured by UV-Vis spectrophotometry [31]. First, standard solution of 100 mg·L^−1^ of benzoic acid was prepared by direct dissolution in deionized water, and 50 mL benzoic acid solutions of 0, 4, 8, 12, 16, and 20 mg·L^−1^ were obtained using the standard solution. Second, the absorbance of different concentrations of benzoic acid solutions was determined with a UV-Vis spectrophotometer (UV-1800) at 227 nm, and the calibration curve was drawn. The absorbance of solution samples was determined at 227 nm according to the absorbance and calibration curve to calculate the concentration of benzoic acid in the solution samples. The concentrations of salicylic acid in the solution samples were measured similarly to benzoic acid. First, standard solution of 100 mg·L^−1^ salicylic acid was prepared by direct dissolution in deionized water, and 50 mL salicylic acid solutions of 0, 4, 8, 12, 16, and 20 mg·L^−1^ were prepared by the standard solution. Second, the absorbance of different concentrations of the salicylic acid solutions was determined with a UV-Vis spectrophotometer (UV-1800) at 297 nm, and the calibration curve was drawn. The absorbance of solution samples was determined at 297 nm according to the absorbance and calibration curve to calculate the concentration of benzoic acid in the solution samples.

The concentrations of ferulic acid, gallic acid, and tannic acid were measured by FeCl_3_–K_3_(Fe(CN)_6_) colorimetry [32,33]. First, 0, 0.5, 1, 1.5, 2.0, and 2.5 mL of ferulic acid solution at the concentration of 20 mg·L^−1^ were put in the corresponding test tubes, and then the solution in each test tube of up to 2.5 mL was made with deionized water. Second, 2.5 mL of 100% ethanol, 2 mL of 0.3% sodium dodecyl sulfate, and 1 mL of 0.6% FeCl_3_–0.9% K_3_(Fe(CN)_6_) (volume ratio 1:0.9) mixture were added into every tube, and then the tubes were placed in the dark for 5 min. Third, 17 mL HCl (0.1 M) was added into each test tube, the tubes were shaken to allow complete reaction of the solution, and then the absorbance of the solution was measured 20 min later by UV-Vis spectrophotometer (UV-1800) at 720 nm. The calibration curve was drawn according to the absorbance and corresponding ferulic acid concentrations. The absorbance of ferulic acid in the solution samples (2.5 mL taken from each repeat) were also measured following the second and third steps noted above, and the concentrations of ferulic acid in the solution samples were calculated according to the calibration curve. The concentrations of gallic acid and tannic acid in solution samples were measured similarly to ferulic acid.

### 2.4. Statistic Analysis

Data were analyzed by the statistical software SPSS 18.0 (International Business Machines Corporation). Data analysis was subjected to one-way analysis of using variance (ANOVA). Significant differences between the means were tested using Duncan’s multiple range test at 95% confidence.

## 3. Results

### 3.1. Plant Morphology and Growth Characteristics

Leaf area of lettuce was highest under W treatment, followed by 8R1B treatment, while leaf area under 2R1B and 4R1B treatment was lower than other treatments (Table 2). There were no significant differences in shoot fresh weight and leaf area among different treatments, but the highest values were observed under W. Root fresh weight was highest under 8R1B treatment, and there was no significant difference among other treatments. Shoot dry weight was also highest under W treatment, but there was no significant difference between W and 8R1B. Additionally, shoot dry weight under other treatments was remarkably lower than W. There was no significant difference in root dry weight among different treatments. Lettuce leaves were long and narrow under white light, and compact under red and blue light (Figure 2).

### 3.2. Leaf Photosynthesis

As shown in Figure 3, leaf photosynthetic rate (*P*n) of lettuce was the lowest under W treatment; it was higher than W under all treatments of red and blue light. *P*n was highest under 3R1B, and it was 142.2% higher than that of W. In addition, 2R1B, 4R1B, and 8R1B increased *P*n by 75.6%, 33.3%, and 104.9%, respectively, compared to W.

### 3.3. Root Morphology

The total root length under 8R1B was the highest, which was 29.8% higher than that under W treatment (Table 3). There was no significant difference in total root length under 2R1B and 3R1B compared to W, and 4R1B was significantly lower than that under W. The results of root surface area were consistent with that of total root length; it was highest under 8R1B and lowest under 4R1B. Root volume under 8R1B was the highest, which was 36.1% higher than that under W. It was significantly lower under 2R1B, 3R1B, and 4R1B than under W.

### 3.4. Root Activity

Root activity of 4R1B treatment was the highest, which was 68.6% higher than that under W, and followed by 3R1B (Figure 4). There was lowest in root activity under 8R1B, which was 33.5% lower than that under W, and no significant difference was observed in root activity under 2R1B and 3R1B treatment compared to W.

### 3.5. Organic Carbon Content

The total organic carbon (TOC) content in nutrient solution under 4R1B was significantly higher than that of W, but there was no significant difference between other treatments and W (Table 4). Except for 2R1B, TOC content/shoot dry weight of other treatments were significantly higher than that of W, and 3R1B, 4R1B, and 8R1B increased by 60.35%, 132.5%, and 56.7%, respectively, compared with W. TOC content/root dry weight was the highest under 4R1B, which was significantly higher than W, and no significant difference was observed between other treatments and W. Similarly, the highest TOC content/root surface area was found in 4R1B treatment, and no significant difference was found between other treatments and W.

### 3.6. Autotoxin Contents

#### 3.6.1. Effect of Light Quality on Five Autotoxins Concentration in Root Exudates of Hydroponic Lettuce

Benzoic acid concentration was the highest under W treatment. Benzoic acid concentration under 2R1B, 3R1B, and 4R1B were all significantly lower than that under W. Moreover, it was the lowest under 8R1B in benzoic acid concentration, which decreased by 42.8% compared to W (Figure 5A). The trend of salicylic acid concentration under all treatments was consistent with that of benzoic acid (Figure 5B). All treatments showed no significant difference in ferulic acid concentration (Figure 5C). The gallic acid concentration in 8R1B treatment was the highest, significantly higher than that in W and other treatments, while there was no significant difference among 2R1B, 3R1B, and 4R1B treatment compared to W (Figure 5D). Concerning tannic acid concentration, it was the highest under 8R1B, while there was no significant difference under 8R1B compared with W and other treatments, except for 2R1B (Figure 5E).

#### 3.6.2. Autotoxin Content Based on Root Surface Area

The value of benzoic acid content/root surface area was the highest under W; it was remarkably higher than other red and blue light treatments, and 8R1B was the lowest in the value of benzoic acid content/root surface area (Figure 6A). In descending order, the value of salicylic acid content/root surface area of treatments was W, 4R1B, 2R1B, 3R1B, and 8R1B (Figure 6B). Ferulic acid content/root surface area was the highest under W, while there was no significant difference between W and 2R1B. The ferulic acid content/root surface area of 3R1B, 4R1B, and 8R1B were significantly lower than W, and 8R1B was the lowest (Figure 6C). There was no significant difference in gallic acid content/root surface area under 2R1B and 4R1B compared to W, but 3R1B and 8R1B treatments were significantly lower than that of W (Figure 6D). The value of tannic acid content/root surface area of 4R1B among all treatments was the highest, but there was no significant difference compared with W. However, those under 2R1B and 3R1B were significantly lower than that under W (Figure 6E).

#### 3.6.3. Autotoxin Content Based on Shoot Dry Weight

The content of benzoic acid/shoot dry weight of W treatment was the highest. Compared with W, the content under 2R1B, 3R1B, and 4R1B treatment decreased 17.8%, 27.0%, and 23.8%, respectively (Figure 7A). The maximum reduction of 8R1B was 50.8%. The trend of the value of salicylic acid content/shoot dry weight was similar to that of benzoic acid content/shoot dry weight, and the salicylic acid content/shoot dry weight under 8R1B decreased by 48.4% compared with W (Figure 7B). Ferulic acid content/shoot dry weight under 2R1B was significantly higher than that under W, while no significant difference was observed between other treatments and W (Figure 7C). Gallic acid content/shoot dry weight was the lowest under 3R1B, and there was no significant difference compared to W under all treatments (Figure 7D). In addition, there was no significant difference in tannic acid content/shoot dry weight among all treatments (Figure 7E).

## 4. Discussion

Light not only provides basic energy for plant photosynthesis, but also plays an important role in regulating plant development, morphogenesis, biosynthesis of cell components, and gene expression throughout the plant life cycle [34,35]. In this study, the photosynthetic rate was low under W, while a larger leaf area made the lettuce capture more light energy, so the lettuce had higher yield under W. This may result from the availability of 11.8% far-red light in the W treatment, as the exposure of plants to light with a higher proportion of far-red is reported to promote total dry mass accumulation and stem extension [36,37]. This was demonstrated by a similar result with long and narrow leaves observed in lettuce under white light. In addition, green light in white light may promote the growth of lettuce. Previous study of lettuce has shown that adding green light to red and blue light significantly increased both shoot fresh and dry weight, because of the ability of green light to penetrate the folded layers of lettuce leaves [38,39]. In our study, a regular change in photosynthetic rate with an increased proportion of red light was not observed. This result is not consistent with the previous research [25], which might be due to different settings of growth environmental factors (light quality, light intensity, and temperature). Furthermore, the lettuce shape is more compact under red and blue light.

Hogewoning et al. [40] noted that cucumber grown under the artificial solar spectrum (AS) with a considerable amount of far-red light had a larger leaf area and longer hypocotyls than that grown under a high-pressure sodium lamp (HPS) and fluorescent tubes. R:FR ratios have an effect on plant morphogenesis, with a general trend of taller plants and longer petioles associated with lower R:FR ratios [41,42]. This phenomenon may help explain the results that the shoot fresh weight and dry weight were larger under W, while the root fresh weight and dry weight had no significant changes compared with other treatments. Previous research indicated that red light was conducive to the synthesis of chlorophyll in the leaves [43], and the combination of red and blue light increased the light utilization efficiency of the light system [44]. Therefore, although the lettuce yield under 2R1B, 3R1B, and 4R1B in this experiment was lower than that under W, when the percentage of red light increased to 88.9%, the yield of lettuce increased remarkably.

Roots are the most important organs of plants, which efficiently obtain nutrients and water from the surrounding environment and secrete organic and inorganic substances, namely root exudates [45]. In the present study, the total root length, root surface area, and root volume were decreased when the proportion of red light increased from 75% (3R1B) to 80% (4R1B), while they were significantly increased when the proportion of red light increased to 88.9% (8R1B). This may be related to the secretion of organic carbon in lettuce roots for the TOC contents. TOC content/shoot dry weight, TOC content/root dry weight, and TOC content/root surface area were increased when the proportion of red light increased from 75% (3R1B) to 80% (4R1B), while they were significantly decreased when the proportion of red light increased to 88.9% (8R1B). The results showed that 4R1B treatment was bad for root morphogenesis of lettuce, not only because of the low photosynthetic rate, but also because plants removed more carbohydrates into the nutrient solution through the root, which was one of the reasons for the low yield under 4R1B treatment. The mechanism of organic carbon secretion responses to red and blue light was not clear; little research has been done in this field. We speculated that the results described above may be mainly related to root activity of lettuce, because the trend of root activity under five treatments was highly consistent with the trend of TOC contents, TOC content/shoot dry weight, TOC content/root dry weight, and TOC content/root surface area. Therefore, we concluded that high root activity increases the secretion of organic carbon in plants (4R1B), while low root activity can reduce the secretion of organic carbon (8R1B). This finding was different from the previous insights about higher root activity making plants absorb more water and nutrients [46]. However, more water and nutrient absorption mediated by root activity was based on the higher photosynthetic rate and more assimilation power to convert the absorbed water and nutrients into carbohydrates [47]. On the contrary, when the photosynthetic capacity of leaves is insufficient, the vigorous root activity will increase the secretion of photosynthetic products, excluding more carbohydrates from the plants and reducing the yield.

Autotoxin is an important factor inducing continuous cropping obstacles [48]. The issue that accumulation of autotoxins in rhizosphere exists not only in soil cultivation but also in hydroponic cultivation of protected horticulture. Lee et al. [8] identified the presence of organic acids in the nutrient solution of lettuce and found that the content of phenolic acid like benzoic acid in the nutrient solution also increased with longer time, while the growth of shoot and root were inhibited. Previous studies have indicated that autotoxins are mainly phenolic acids, including benzoic acid, 4-hydroxy-benzoic acid, cinnamic acid, salicylic acid, ferulic acid, gallic acid, and tannic acid [10,49,50,51]. Our previous results confirmed that benzoic acid, salicylic acid, ferulic acid, gallic acid, and tannic acid are all present in root exudates from lettuce [23]. In this study, light quality had different effects on the secretion of different autotoxins. The value of autotoxin content/root surface area and autotoxin content/shoot dry weight under white light were generally higher than other treatments, such as benzoic acid content/root surface area and salicylic acid content/shoot dry weight. This indicated that the lettuce with the same root surface area or same accumulation in shoot dry weight secreted more autotoxins into the nutrient solution under white light, especially benzoic acid and salicylic acid. Autotoxins led to the disorder of plant metabolism, the inhibition of water and nutrients absorption, and to the accumulation of reactive oxygen species in plants [52,53]. The membrane structure of the plant was then destroyed by autotoxicity, the permeability of membrane increased, the electrolyte inside the cell extravasated, and, in severe cases, the root tip cell would die [53]. In addition, autotoxins not only directly affected the root system of lettuce, but also indirectly inhibited the photosystem function and photosynthesis by destroying the structure of chloroplast [54]. This is also the reason for the low photosynthetic rate and root biomass of lettuce under white light. 

Some insights into the effects of light quality on plant secondary metabolism can be gained from agronomic studies exploring plant–plant communications and competitions. Compared with white light, a different ratio of red to blue effectively reduced the secretion of benzoic acid and salicylic acid in lettuce roots, and reduced the ability for secretion of benzoic acid and salicylic acid by per unit root surface area and accumulation by shoot dry weight. This may be the result of 11.8% far-red in white light. When plants were shaded by neighboring plants, changes of light quality in red:far-red ratio was perceived by phytochrome and other informational photoreceptors [55,56,57]. This signal of light quality change caused the variation of secondary metabolism in plants, thus changing the root structure and the profile of root exudates, increasing some compounds with potential allelopathy [58,59,60]. Furthermore, 22.0% of green light in white light may be one of the reasons for the high secretion of benzoic acid and salicylic acid, because adding green light to red and blue light promoted secondary metabolism in lettuce by regulating related gene expression and enzyme activities [61]. In addition, compared with the low proportion of blue light (8R1B), the secretion of benzoic acid and salicylic acid was increased in a high proportion of blue light (2R1B, 3R1B, 4R1B). This is, perhaps, because enhanced blue light ratio may strongly increase the biosynthesis of epidermal flavonoids and phenolic compounds [62,63]. Furthermore, red light promoted the accumulation of carbohydrates in plants [25,34], which may lead to a decrease in the secretion of autotoxins under a high proportion of red light (8R1B). Besides, changes in light quality by photoreceptor mediation also brought about a variety in patterns of hormone distribution, such as jasmonic acid in *Lotus japonicus* nodulation [64]. This effect resulted in within-plant competition for resources (e.g., competition for carbohydrates between shoots and roots), and ultimately leading to changes in the quality and quantity of compounds that plants emit or secrete to regulate the belowground interactions of plants [65,66]. Thus, light quality perceived by the shoots could affect the profile of root exudates by regulating the secondary metabolism and hormone distribution pattern in plants, but this possibility has received little experimental attention.

In our experiment, we confirmed that different red to blue ratios of LED light decreased the secretion of benzoic acid and salicylic acid, and it was the most obvious under 8R1B treatment. However, the decrease was not linear with the proportion of red or blue light, which may be related to the complexity of the influence of light quality on plant growth. Under 8R1B treatment, the lettuce with the same root surface area or same accumulation in shoot dry weight generally secreted less autotoxins into the nutrient solution, especially benzoic acid and salicylic acid. In addition, less organic carbon was secreted under the treatment. These also explain why this treatment had a higher shoot fresh weight, root fresh weight, total root length, root surface area, and root volume, compared to other treatments. Therefore, 8R1B reduced the secretion of organic carbon and autotoxins not at the expense of lettuce yield. Moreover, we propose a hypothesis that the effect of light quality on root exudates is not only associated with secondary metabolism and hormone distribution in plants, but also related to transportation and secretion of exudates in roots. The current study, however, is very limited in this aspect and further research is needed.

## 5. Conclusions

Overall, the lettuce had higher yields under W and 8R1B treatments at 200 μmol·m^−2^·s^−1^ light intensity in short-term cultivation (15 d). Light quality with different R/B ratios showed pronounced effects on the secretion of organic carbon and autotoxin. Compared with white light, different ratios of red to blue effectively reduced autotoxins secretion, based on the same root surface area or same accumulation in shoot dry weight, especially for benzoic acid and salicylic acid. In addition, 8R1B reduced the secretion of organic carbon and autotoxins not at the expense of lettuce yield. Therefore, we conclude that 8R1B treatment is recommended in the view of reducing the secretion of autotoxins while ensuring the yield of lettuce. This treatment will increase nutrient solution utilization efficiency and reduce the probability of continuous cropping obstacles in plant factory.

## Figures and Tables

**Figure 1 plants-09-01542-f001:**
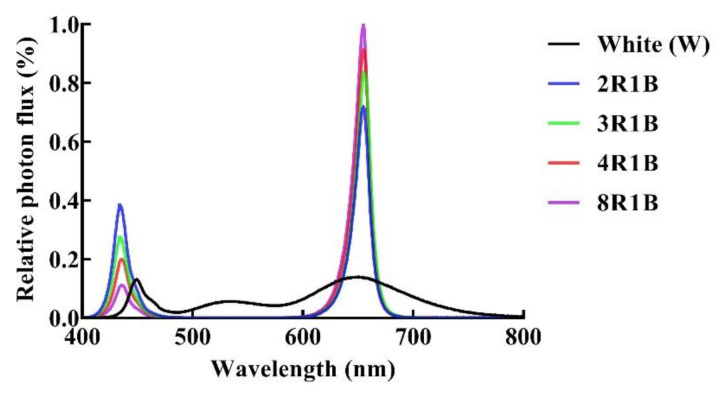
Relative spectral distributions of the red and blue LED light used in this study.

**Figure 2 plants-09-01542-f002:**
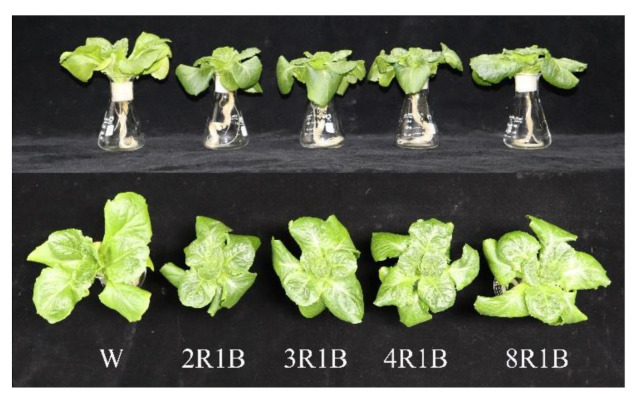
Morphology of lettuce plants grown under different R/B ratios of light intensity (2R1B, 3R1B, 4R1B, and 8R1B) and white LED (W, control) treatments on photosynthesis of hydroponic lettuce.

**Figure 3 plants-09-01542-f003:**
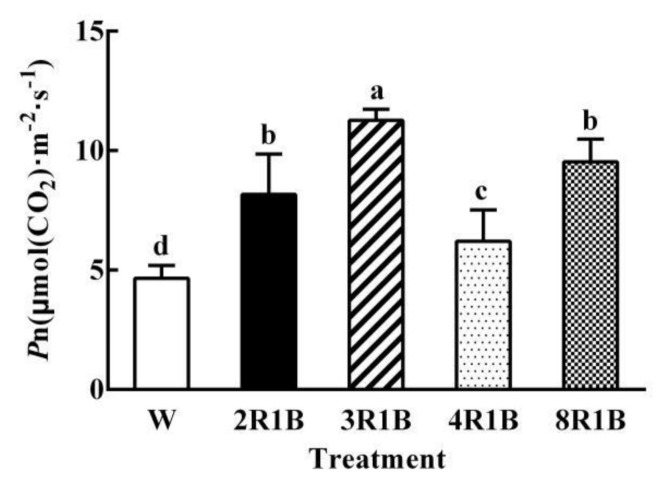
Effects of different R/B ratios of light intensity (2R1B, 3R1B, 4R1B, and 8R1B) and white LED (W, control) treatments on photosynthesis of hydroponic lettuce. Notes: different letters for the same parameter indicate significant difference at the 5% level, according to the Duncan’s test (*n* = 4). The bars represent the standard errors.

**Figure 4 plants-09-01542-f004:**
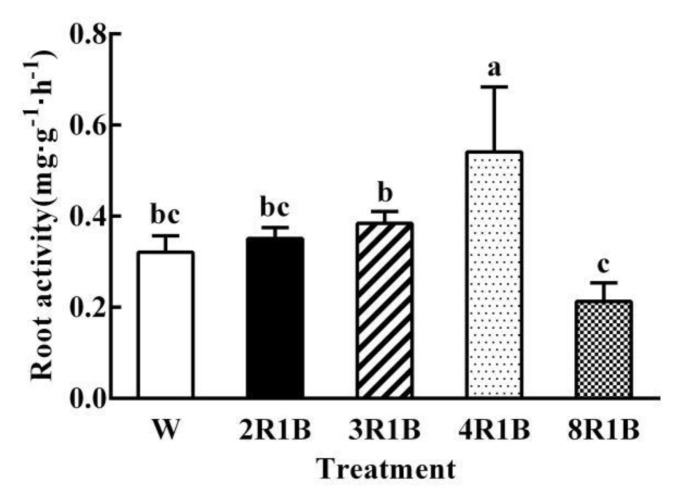
Effects of different R/B ratios of light intensity (2R1B, 3R1B, 4R1B, and 8R1B) and white LED (W, control) treatments on root activity of hydroponic lettuce. Notes: different letters for the same parameter indicate significant difference at the 5% level, according to the Duncan’s test (*n* = 4); the bars represent the standard errors.

**Figure 5 plants-09-01542-f005:**
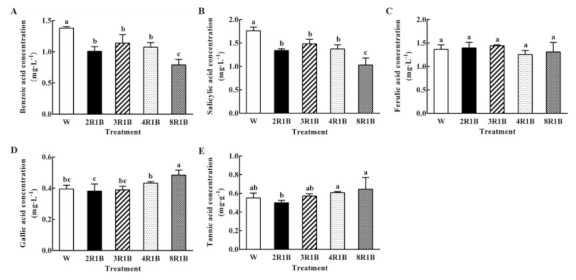
Effects of different R/B ratios of light intensity (2R1B, 3R1B, 4R1B, and 8R1B) and white LED (W, control) treatments on benzoic acid (**A**), salicylic acid (**B**), ferulic acid (**C**), gallic acid (**D**), and tannic acid (**E**) concentration in root exudates of hydroponic lettuce. Notes: different letters for the same parameter indicate significant difference at the 5% level, according to the Duncan’s test (*n* = 4); the bars represent the standard errors.

**Figure 6 plants-09-01542-f006:**
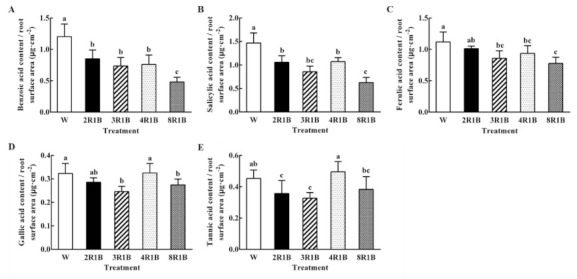
Effects of different R/B ratios of light intensity (2R1B, 3R1B, 4R1B, and 8R1B) and white LED (W, control) treatments on benzoic acid (**A**), salicylic acid (**B**), ferulic acid (**C**), gallic acid (**D**), and tannic acid (**E**) content, secreted from the root surface area of hydroponic lettuce. Notes: different letters for the same parameter indicate significant difference at the 5% level, according to the Duncan’s test (*n* = 4); the bars represent the standard errors.

**Figure 7 plants-09-01542-f007:**
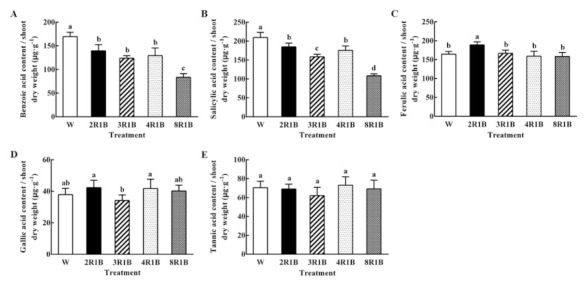
Effects of different R/B ratios of light intensity (2R1B, 3R1B, 4R1B, and 8R1B) and white LED (W, control) treatments on benzoic acid (**A**), salicylic acid (**B**), ferulic acid (**C**), gallic acid (**D**), and tannic acid (**E**) content, secreted by unit shoot dry weight accumulation in lettuce. Notes: different letters for the same parameter indicate significant difference at the 5% level, according to the Duncan’s test (*n* = 4); the bars represent the standard errors.

**Table 1 plants-09-01542-t001:** Summary of the spectral qualities tested for the white light (WL) and the combination of red and blue light (RB) (μmol·m^−2^·s^−1^).

Treatments	PFD	Blue (400–500 nm)	Green (500–600 nm)	Red (600–700 nm)	Far-Red (700–800 nm)
White (W)	200	31.6 (15.8%)	43.9 (22.0%)	100.6 (50.3%)	23.5 (11.8%)
2R1B		66.7 (33.3%)	－	133.3 (66.7%)	－
3R1B		50.0 (25.0%)	－	150.0 (75.0%)	－
4R1B		40.0 (20.0%)	－	160.0 (80.0%)	－
8R1B		22.2 (11.1%)	－	177.8 (88.9%)	－

PFD: photon flux density.

**Table 2 plants-09-01542-t002:** Effects of different R/B ratios of light intensity (2R1B, 3R1B, 4R1B, and 8R1B) and white LED (W, control) treatments on the leaf area, shoot fresh weight, root fresh weight, shoot dry weight, and root dry weight of hydroponic lettuce.

Treatments	Leaf Area (cm^2^)	Shoot Fresh Weight (g)	Root Fresh Weight (g)	Shoot Dry Weight (g)	Root Dry Weight (g)
W	287.8 ± 39.6 a	14.5 ± 2.49 a	2.36 ± 0.75 ab	0.80 ± 0.11 a	0.124 ± 0.03 a
2R1B	182.9 ± 24.0 c	9.1 ± 1.25 c	1.89 ± 0.42 ab	0.59 ± 0.06 b	0.093 ± 0.02 a
3R1B	200.8 ± 23.7b c	10.0 ± 1.07 bc	1.93 ± 0.75 ab	0.58 ± 0.03 b	0.094 ± 0.03 a
4R1B	184.5 ± 25.3 c	9.2 ± 1.71 c	1.75 ± 0.20 b	0.53 ± 0.09 b	0.093 ± 0.01 a
8R1B	235.6 ± 45.9 b	12.4 ± 2.31 ab	2.71 ± 0.44 a	0.68 ± 0.11 ab	0.125 ± 0.02 a

Notes: values were means of four replicates ± SD; means within rows followed by different letters are significantly different according to Duncan’s multiple test (*p* ≤ 0.05).

**Table 3 plants-09-01542-t003:** Effects of different R/B ratios of light intensity (2R1B, 3R1B, 4R1B, and 8R1B) and white LED (W, control) treatments on the total root length, root surface area, and root volume of hydroponic lettuce.

Treatments	Total Root Length (cm·plant^−1^)	Root Surface Area (cm^2^·plant^−1^)	Root Volume (cm^3^·plant^−1^)
W	2578.0 ± 451.8 b	220.6 ± 31.8 b	1.47 ± 0.11 b
2R1B	2098.2 ± 54.3 bc	169.6 ± 5.5 bc	1.03 ± 0.09 c
3R1B	2240.9 ± 705.2 bc	180.2 ± 52.2 bc	1.16 ± 0.30 c
4R1B	1616.6 ± 151.5 c	152.2 ± 1.7 c	1.11 ± 0.15 c
8R1B	3345.9 ± 249.8 a	283.6 ± 3.3 a	2.00 ± 0.07 a

Notes: values were means of four replicates ± SD; means within rows followed by different letters are significantly different according to Duncan’s multiple test (*p* ≤ 0.05).

**Table 4 plants-09-01542-t004:** Effects of different R/B ratios of light intensity (2R1B, 3R1B, 4R1B, and 8R1B) and white LED (W, control) treatments on total organic carbon (TOC) secretion of hydroponic lettuce.

Treatments	TOC Content (mg)	TOC Content/Shoot Dry Weight (mg·g^−1^)	TOC Content/Root Dry Weight (mg·g^−1^)	TOC Content/Root Surface Area (mg·cm^−2^)
W	27.49 ± 0.5 4b	39.58 ± 2.56 c	265.2 ± 60.5b	0.157 ± 0.05 bc
2R1B	33.12 ± 0.99 ab	53.57 ± 1.19 bc	303.9 ± 63.6b	0.193 ± 0.01 b
3R1B	37.45 ± 2.09 ab	63.44 ± 2.48 b	337.8 ± 26.8b	0.150 ± 0.01 bc
4R1B	45.19 ± 6.73 a	92.03 ± 9.34 a	505.6 ± 80.9a	0.335 ± 0.02 a
8R1B	29.53 ± 11.70 b	62.01 ± 11.25 b	279.1 ± 98.8b	0.122 ± 0.04 c

Notes: values were means of four replicates ± SD; means within rows followed by different letters are significantly different according to Duncan’s multiple test (*p* ≤ 0.05).

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
