# Peer review of "Light Quality Affected the Growth and Root Organic Carbon and Autotoxin Secretions of Hydroponic Lettuce"

_plants, 2020, doi:10.3390/plants9111542_

Round 1

Reviewer 1 Report

Please see attached file for the comments 

Reviewer 2 Report

REVIEW OF THE ARTICLE BY CHENGBO ZHOU ET AL. ENTITLED ‘LIGHT QUALITY AFFECTED THE GROWTH AND ROOT ORGANIC 3 CARBON AND AUTOTOXIN SECRETIONS OF HYDROPONIC LETTUCE’ (plants-973541)

Zhou et al. evaluated the effect of light quality on the morphology, physiological parameters and accumulation of benzoic, gallic, tannic, ferulic,  and salicylic acids in roots in the Lactuca sativa L. They used different proportions of bule and red light as well as white light as a control The text is well-written. Although the issue of L. sativa biology, especially, the effect of red and blue light, is widely studied, the Authors demonstrated new data on plant-specific metabolites accumulation. Introduction is sufficient, most of experimental procedures are correctly described. Results description is also good. At the same time there are some drawbacks. Thus, the article will be acceptable after a revision in accordance with my suggestions. Please, find them below.

  1. I detected only one significant drawback: the description of light quality is to confusing.  The Authors used different proportions of bule and red light as well as white light as a control, however what does this ration mean: whether it ratio of light quant or total energy, or other... It should be clearly indicated in the abstract and in the main body of the text.
  2. I have also one important conceptual question. The Authors says: "...five autotoxins secretion (benzoic acid, salicylic acid, ferulic acid, gallic acid and tannic acid)..." (l. 93-94). Listed compounds are not by-products, they are synthesized through special pathways in plant tissues and play very important role. In fact, salicylic acid is a plant hormone modulating, first af all, immune response to pathogens. Gallic acid and tannic acids also play important role in plant growth and development. Thus, I strongly disagree, that these compounds should be considered as 'toxins'. Moreover, taking into account lower biomass yield after non-white light treatment, decreasing of salicylic acid and other acids might be unfavorable for plant. In addition, decreasing of salicylic acid content might affect possible immune response to possible pathogens making the plants less resistant. It should be discussed.
  3. There are some specific comments.

-Please, unify '/' and negative powers through the text.

-Please indicate, whether the white light was cold-white or warm-white.

-l. 102. Atmospheric CO2 concentration is μmol per mol of what?

-l. 106. Please, provide the reference for Hoagland’s solution.

-Figure 1. Please, provide the procedure of spectra obtaining.

-l. 142. 0.4% (wt/v)?

-l. 142. Please, indicate 'phosphate buffer' (pH, concentration, was it Na or K--phosphate).

-l. 162-169. The description of the procedure of spectrophotometric acids determination is confusing. Please, describe, whether other aromatic compounds in the solution might affect absorbance in the UV range. Plants exhibit a wide range of aromatic compounds from nucleic acids, proteins and nucleotides to phenolic compounds. Why are the Authors shure, that they measured strictly the concentrations of Benzoic acid and salicylic acid? Moreover, absorbance spectra of these two compounds are overlapped. Therefore, it is necessary to determine optical densities at two wavelengths and use the system of two linear equation for determination of the concentration of each compound. Please describe it, and it would be nice to see this system of equations, e.g. for a standard cuvette with the optical path length of 1 cm.

-l. 172. What is the reason of 99% ethanol using. For me, it is very strange. As a rule there are standard labour 96% ethanol and absolute 100% ethanol?

-l. 180 and through the text. 'standard curve' should be 'calibration curve'.

-Please, describe symbols W, 2R1B, 4R1B, and 8R1B in the legends of figures and tables.

-Please, describe, what is the length of the root. As wast majority of plants, L .sativa exhibit branched root system. Is it the length of the main root? Whas the degree of root ramification different under different experimental conditions.

-l. 297. " This result may result" - please, rephrase.

-It is also would be interesting to calculate and compare surface-to-volume ratio for roots, since this important parameter says about activity of nutrients absorption and whole metabolic activity, rather than surface and volume separately. I think, it will enhance results and discussion.

-l. 349-356. Reverences are required.

-l. 3556-356. Difference in CO2 flow might be not only due to decreasing of photosynthesis rate, but also due to different balance between respiration and photosynthesis.

-l. 360-376. The difference in the obtained results also might be explain by the fact, that white light also contain a green component. Although green light is not absorbed by photosynthetic pigments, it is detected by cells and has physiological effects on plants. Please, discuss it also.

-Please, discuss the results on photosynthesis and plant morphology with the references on the studies done mainly on L. sativa. There are many such data, eg ref 27, 28, 29, 30.

-I cannot find some references in the open sources, e.g. 27, 28.

Reviewer 3 Report

The manuscript is interesting and the authors have tried to solve a problem related to lettuce production in hydroponic. I  appreciate this idea. Overall, this MS is written well with sufficient information in the introduction and discussion. I have a few suggestions to improve the manuscript: 

 It is good to show a correlation between autotoxin secretion and growth rate of lettuce? since lettuce is fresh edible food, so that, flavor and crispiness analysis are important as well as a flavonoid and phenolic acid content of lettuce and their correlation with autotoxic secretion. 

It is also desirable to show how lettuce root exudates affect nutrient availability in the solution? Moreover, the nutrient volume is very small only 4 L and only 4 plants were considered for data collection, BUT  this type of study, it is always good to have more plants >50 plants in the study per treatment and 10 plants for the analysis out of 50 plants. 

What hydroponic method was used during the study: NFT, Ebb and Flow or others? 

How the physically RB light combination was? was the R and B light chip in the same light stick or 100% red light stick, 100% blue light stick individual then arranged them 4R and 1B for treatment setting? I mean is it tube light or panel light? I went through the company page but didn't get the used light. The light used in this study, it seems it is not horticultural light? the company makes household light.  Hope the authors will make me understand. 

How about analysis of other minerals like: P, Na, Ca, S etc... in the hydroponic solution? because it's important to make a high-quality manuscript. 

Important point: 

Please change the figure 1 color of each treatment. It is difficult to identify the individual  light combination

Line 100: was the seedling grown in the same plant factory at the same temperature? 

line 102: it should be co2 in a plant factory not atmospheric? 

line : 293-299: it is not convincing explanation: how FR light decrease pn and increase leaf area? if W light content FR light, hows the percentage? 

line 321-322: how plant roots remove carbohydrate into the solution? 

 line 354-356: 'In addition, autotoxins
354 not only directly affected the root system of lettuce, but also indirectly inhibited the photosystem function and photosynthesis by inhibiting the synthesis of chloroplast. This is also the reason for the low photosynthetic rate and root biomass of lettuce under white light' need references

line 357-359: 'This experiment was completed after 15 d of lettuce planting, so we speculated that the growth of lettuce would be  significantly inhibited after 15 d or next cultivation period under white light because of the accumulation of autotoxins in nutrient solution' this statement is a hypothesis. its not scientific statemetn until performing further research

' Moreover, we propose a hypothesis that the effect of light quality on root exudates is not only associated with secondary metabolism and hormone distribution in plants, but also related to transportation and secretion of exudates in roots. But, the current study is very limited in this aspect
 and further study is needed'. this is good ending. appreciated

conclusion :  OK 

Round 2

Reviewer 2 Report

REVIEW OF THE ARTICLE BY CHENGBO ZHOU ET AL. ENTITLED ‘LIGHT QUALITY AFFECTED THE GROWTH AND ROOT ORGANIC 3 CARBON AND AUTOTOXIN SECRETIONS OF HYDROPONIC LETTUCE’ (plants-973541)

The Authors significantly improved the text, and I am almost completely satisfied. But there are still several formal issues.

-Although the Authors describe light quality more detail, I think it remains unclear. First, light quality in μmol m-2 s-1 is not light intensity, but photon flux density, whereas intensity is energy based units! What did you actually use? intensity or PFD? Secondly, since R/B = 2, 2R1B; R/B = 3,3R1B; R/B = 4, 4R1B; and R/B = 8, 8R1B are ratios, are these actually actually R/B = 2.2; R/B 17 = 3.3; R/B = 4.4; and R/B = 8.8?

-The procedure of PFD or light intensity determination should be specified.

-l. 98. Please, check the formula. Is it actually ammonium molybdate?

l. 104 and l. 109: which procedure of light intensity determination is correct?

-I am not sure about Journal's rules, but should the references to non-English sources be marked is 'In Chinese'?
